# Spatio-temporal distribution and socioeconomic inequality of low birthweight rate in China from 1992 to 2021 and its predictions to 2030

Chengyue Li[1], Lixia Lei[2], Yingying Li[1]*

1 Institute of Physical Education, Xinjiang Normal University, Urumqi, Xinjiang, China, 2 Department of Oncology, Urumqi Chinese Medicine Hospital, Urumqi, China

* 1301073231@qq.com

**Data Availability Statement:** All relevant data are within the article and its Supporting Information files.

## Abstract

This paper aims to investigate the trend, spatio-temporal distribution, and socioeconomic inequality of the low birthweight rate (LBWR) in China from 1992 to 2021 and to project the LBWR to 2030. We performed a secondary analysis of data from the China Health Statistics Yearbook. LBWR refers to the ratio of the number of infants born with a birth weight less than 2,500 grams to the number of live births in a given year. We used joinpoint regression models to estimate LBWR trends from 1992 to 2021 for the whole country and from 2002 to 2021 for the three regions (eastern, central, and western regions) and each province. The slope index of inequality (SII) and relative index of inequality (RII) were calculated for each year from 2002 to 2021 based on provincial data. LBWR increased from 2.52% (1992) to 3.70% (2021), and the average annual percentage change (AAPC) (95% confidence interval [CI]) was 1.35% (0.22%, 2.49%) in China. The overall LBWR from 2002 to 2021 was greatest in the Eastern region, but LBWR had the fastest increase in the Western region, with an AAPC (95% CI) of 3.15% (2.59%, 3.12%). There were spatio-temporal differences in the LBWR and trends between provinces. The SII and RII increased linearly from -0.15 and 0.94 to 0.53 ($B$ = 0.035%, $p$ < 0.001) and 1.16 ($B$ = 0.011, $p$ < 0.01), respectively, over the past 20 years. The results of the ARIAM model showed that the National LBWR will be increasedfrom 3.70% in 2021 to 5.28% in 2030. The LBWRs in the eastern, central and western regions in 2030 will be 4.93%, 6.02% and 5.82%, respectively. National and local governments must prioritize disadvantaged groups to mitigate the rapid prevalence of LBWR, reduce regional disparities, and improve perinatal and infant health and health equity in China.

## Introduction

According to the definition of the World Health Organization (WHO), a living birth infant with a weight <2500 grams was regarded as a low birth weight (LBW) infant [1]. LBW is an

**Funding:** This study was supported by the Natural Science Foundation of Xinjiang Uygur Autonomous (2022D01B108 to YL).

**Competing interests:** The authors have declared that no competing interests exist.

important indicator of the health of a population and a predictor of children's health in the near and distant future. LBW is a significant predictor of perinatal mortality and morbidity compared to normal birth weight infants [2]; increases the prevalence of stunting and intellectual and executive functioning deficits in childhood [3,4]; is associated with decreased physical and cognitive functions, intelligence, lung functions, educational attainment, and employment rates in adulthood [5–8]; and increases the risk of chronic noncommunicable diseases, coronary heart disease, and diabetes in adulthood, as well as metabolic syndrome [9–12].

To improve maternal, infant and child nutrition, the World Health Assembly (WHA) set the global nutrition goal of "reducing global low birthweight rate (LBWR) by 30% by 2030" [3]. Only one global effort pooled LBWR from 148 countries worldwide revealed that the global LBWR decreased from the beginning of the 21st century to 2015, with national and regional differences [13]. Some countries or territories, such as West Africa [14], Japan [15,16], Peru [17], Ireland [18], the United States [19,20], Australia [21,22], and Mexico [23], have recently provided updated monitoring data. Overall, there has been insufficient evidence in recent years, especially in developing countries, including China. Previous studies have shown that the LBWR in China was less than 5% in 2015, which is much lower than the global average (14.3%) [13]; however, increasing trends have been observed, and recent trends are unknown. Given that China has approximately 1/5 of the world's population and a large number of newborns, the achievement of nutritional goals in China has great significance for the achievement of global nutritional goals.

LBWR varies not only between countries but also within countries [17,22]. Only one study from China from a decade ago investigated this spatial distribution. It is worth noting that this study included data from only approximately half of the provinces in China [24]. However, no study has evaluated the spatio-temporal distribution characteristics of the LBWR in China in recent years (with differences across provinces). Although the evaluative indicators used are different, socioeconomic status (SES) is associated with LBWR [16,17,19,21–23]. There is lack of studies quantifying socioeconomic inequality in LBWR in China, and it is not clear how this socioeconomic inequality changes over time. High-quality and fine-grained data are needed to provide evidence for the development and implementation of LBWR reduction interventions globally, especially in low- and middle-income countries. However, national interventions are not sufficient to achieve intervention outcomes in countries that have large landmasses with large and diverse populations (e.g., China). Therefore, provincial survey data can provide unique information for identifying not only the limitations of current policy implementation but also the areas prioritized for intervention.

Therefore, we used national survey data from 1992 to 2021 and provincial survey data from 2002 to 2021 to estimate (1) national and provincial trends of LBWR, (2) the spatio-temporal distribution of the provincial LBWR, (3) socioeconomic inequalities of LBWR and relevant trends, and (4) the predicated results of LBWR from 2022 to 2030.

## Materials and methods

### Data sources

National LBWRs from 1992 to 2021 and provincial LBWRs from 2002 to 2021 were obtained from the China Health Statistical Yearbook (2003–2022) [25]. The gross domestic product (GDP) per capita and population by province from 2002 to 2021 were obtained from the National Bureau of Statistics of China [26]. LBWR refers to the ratio of the number of infants born with a birth weight less than 2,500 grams to the number of live births in a given year [25].

## Statistical analysis

Jointpoint regression was performed using the JointPoint Regression Program 4.9.1.0 (National Cancer Institute, Bethesda, MD, USA) to analyze LBWR trends. The maximum number of joinpoints was determined according to the default options of the system, the number of joinpoints given by the model was subsequently determined, and the annual percentage change (APC) and average APC (AAPC) (the APC of the model without joinpoints is the AAPC) and their 95% confidence intervals (CIs) were estimated for the nation from 1992 to 2021 and for each province from 2002 to 2021. The calculation process for the APC was divided into two steps: (1) $y = \alpha + \beta x + \varepsilon$, where $y = \ln(LBWR)$, $x = year$, and $\varepsilon$ is the error term; (2) $APC = 100 \times [\exp(\beta) - 1]$. The weighted Bayesian information criterion (WBIC) was used to assess the quality of the model fit. Lower values of WBIC indicate a better fit.

To investigate the geographical distribution of LBWR, we first divided the 30 provinces, autonomous regions and municipalities into eastern, central, and western regions according to the classification method of the National Bureau of Statistics [26]. The eastern regions included Beijing, Fujian, Guangdong, Hainan, Hebei, Heilongjiang, Jiangsu, Jilin, Liaoning, Shandong, Shanghai, Tianjin and Zhejiang; the central regions included Anhui, Henan, Hubei, Hunan, Jiangxi and Shanxi; and the western regions included Chongqing, Gansu, Guangxi, Guizhou, Neimenggu, Ningxia, Qinghai, Shaanxi, Sichuan, Xinjiang and Yunnan. We used the 2021 population across provinces as a weight for calculating the LBWRs in the three geographic areas for each year. In addition, we used ArcMap 10.8.1 (Esri, Redlands, CA, United States) to construct maps based on provincial LBWRs for 2002 and 2021, as well as maps showing differences in trends represented by AAPC values. Vector files of the maps were obtained from the National Catalogue Service of Geographic Information [27] which was updated in 2021.

The slope index of inequality (SII) and relative index of inequality (RII) were used as indicators of socioeconomic inequality in the LBWR. The SII is a measure of absolute inequality and is the difference between the highest and lowest socioeconomic level subgroups when considering all the other subgroups in the regression (that is, the effect of change in the whole distribution of the population by education or wealth) [28,29]. The SII is an index based on regression estimation, with the x-axis representing the cumulative percentage of the population (ranked from lowest to highest GDP per capita in each province). The values of the x-axis at which the midpoint of each population distribution subgroup was located were used as the relative ranking of the socioeconomic level of the group (values from 0 to 1). The y-axis represented the LBWR, and a linear regression was performed using weighted least squares, with the weights being the percentage of the population in each province and the slope of the regression representing the SII; that is, the absolute value of the LBWR changed as the socioeconomic ranking rose from 0 to 1. The RII is a measure of relative inequality, that is, the ratio of the health status of the highest to the lowest socioeconomic level subgroup [28]. It is given by RII = (intercept+slope)/intercept. We calculated the SII and RII and their 95% CIs for each year from 2002 to 2021 using linear regression to estimate trends.

R4.3.2 (R Foundation for Statistical Computing, Vienna, Austria) was used to construct the autoregressive integrated moving average (ARIMA) model for time series forecasting. In ARIMA (p, d, q), the p parameter represents the order of autoregression and is determined by the partial autocorrelation figure (PACF), while the q parameter represents the order of the moving average and is determined by the autocorrelation figure (ACF). The d parameter represents the number of differentials required to achieve smoothness of the sequences. Using the Ljung-BoxQ test, the residual sequence was considered white noise if the p value was greater than 0.05. We also selected the best model by the Bayesian information criterion (BIC). In this study, ARIMA models were fitted using national data from 1992 to 2021 and regional data

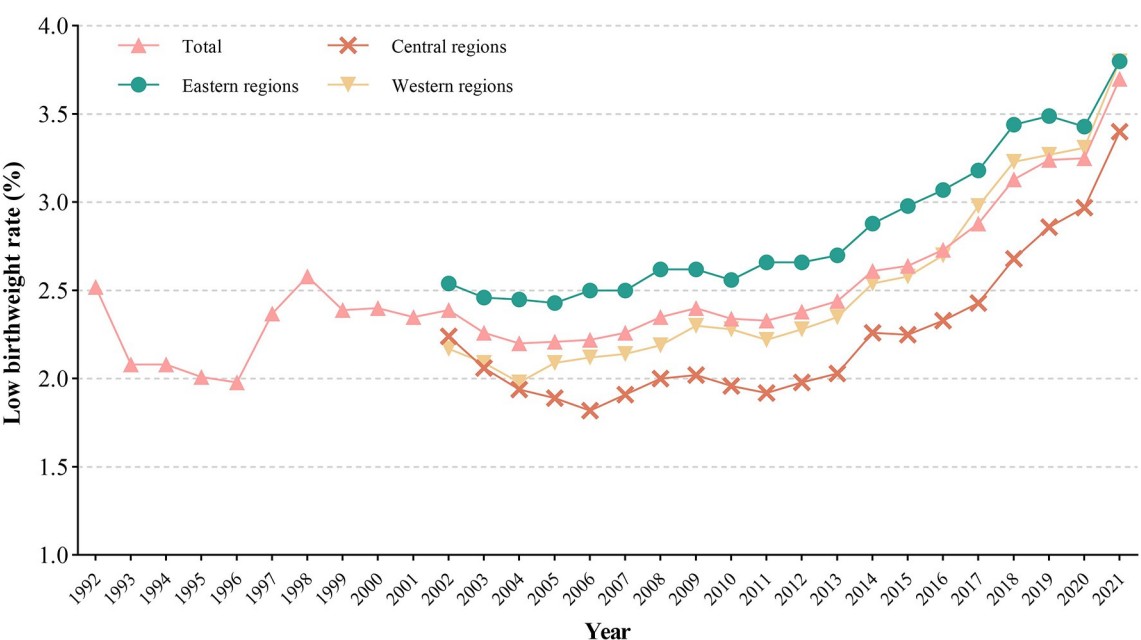

**Fig 1. Trend of the Chinese low birthweight rate from 1991 to 2020 by area.**

from 2002 to 2021 to predict the national and three geographic regions' LBWRs from 2022 to 2030. The level of statistical significance was set at 0.05.

## Results

### Trend of LBWR

LBWR in China increased from 2.52% to 3.70% from 1992 to 2021. The LBWRs in the eastern, central and western regions increased from 2.54%, 2.24% and 2.17% in 2002 to 3.80%, 3.40% and 3.80% in 2021, respectively (Fig 1). Table 1 reveals the APC and AAPC of the joinpoint regression. A significant decline in the national LBWR was observed for two periods, 1992 to 1995 and 1998 to 2004, respectively, and significant increases were observed in the other periods (all $p < 0.05$). The AAPC (95% CI) for the national LBWR over the past 30 years was 1.35% (0.22%, 2.49%) ($p < 0.05$). LBWR exhibited the fastest increase from 2002 to 2021 in the western regions with an AAPC of 3.15%, which was significantly greater than that in the eastern (2.25%) and central regions (2.01%). Although the AAPC in the eastern regions was greater than that in the central region, the difference was not significant.

### Trends of LBWR across provinces

Gansu (5.23%) had the largest LBWR in 2002, followed by Hebei (4.34%) and Qinghai (3.96%), and Xinjiang (1.01%) had the lowest LBWR. Fifteen of the 30 provinces had LBWRs between 1% and 2%, and 10 provinces had LBWRs between 2% and 3%. Four provinces had LBWRs of more than 5% in 2021, including Guangxi (5.89%), Hainan (5.61%), Shanghai (5.50%) and Beijing (5.40%) (Fig 2). Fig 3 reveals the geographic distribution of the LBWR and its changes in China from 2002 to 2021. In 2002, provinces with greater LBWR were concentrated in the western regions and along the Bohai Sea regions, and in 2021, they were concentrated in the northwest regions, near the capital, and in some southern provinces. LBWR increased in 25 provinces over the past 20 years. Xinjiang had the fastest increase in LBWR,

**Table 1. Joinpoint regression of low birthweight rates in the nation, eastern, central and western regions.**

| Region | Year | APC (95%CI) | AAPC (95% CI) | AAPC comparison (95% CI) |
|---|---|---|---|---|
| Total | 1992–1995 | -7.37 (-11.51, -3.04) ** | 1.35 (0.22, 2.49) * | |
| | 1995–1998 | 9.64 (0.06, 20.13) * | | |
| | 1998–2004 | -2.18 (-4.16, -0.06) * | | |
| | 2004–2013 | 1.01 (0.01, 2.12) * | | |
| | 2013–2021 | 4.92 (3.88, 5.97) ** | | |
| Eastern regions | 2002–2012 | 0.78 (0.23, 1.33) ** | 2.25 (1.86, 2.65) ** | Eastern regions vs. Central regions: 0.25 (-0.91, 1.40) |
| | 2012–2021 | 3.91 (3.25, 4.58) ** | | |
| Central regions | 2002–2005 | -5.38 (-10.45, -0.02) * | 2.01 (0.93, 3.10) ** | Eastern regions vs. Western regions: -0.90 (-1.59, -0.21) * |
| | 2005–2013 | 0.56 (-0.89, 2.03) | | |
| | 2013–2021 | 6.01 (4.75, 7.29) ** | | |
| Western regions | 2002–2013 | 1.17 (0.50, 1.84) ** | 3.15 (2.59, 3.72) ** | Central regions vs. Western regions: -1.15 (-2.37, 0.07) * |
| | 2013–2021 | 5.94 (4.81, 7.09) ** | | |

APC, annual percentage change; AAPC, average annual percentage change; CI, confidence intervals

*$p < 0.05$

**$p < 0.01$.

with an AAPC (95% CI) = 8.22% (6.58%, 9.89%), increasing from the lowest ranking in the country to the 5th. The second fastest increase was in Tianjin, with an AAPC (95% CI) = 6.62% (3.02%, 10.35%). LBWR decreased in 5 provinces, namely, Gansu, Hebei, Qinghai, Henan, and Liaoning, with the fastest decrease occurring in Gansu (AAPC [95% CI] = -2.61% [-3.47%, -1.74%]). The APC and AAPC for each province are shown in S1 Table.

## Socioeconomic inequality of LBWR

The SII and RII in 2002 were -0.15 and 0.94, respectively, indicating that the lower SES group had a high LBWR. However, the SII and RII in 2021 were 0.53 and 1.16, respectively,

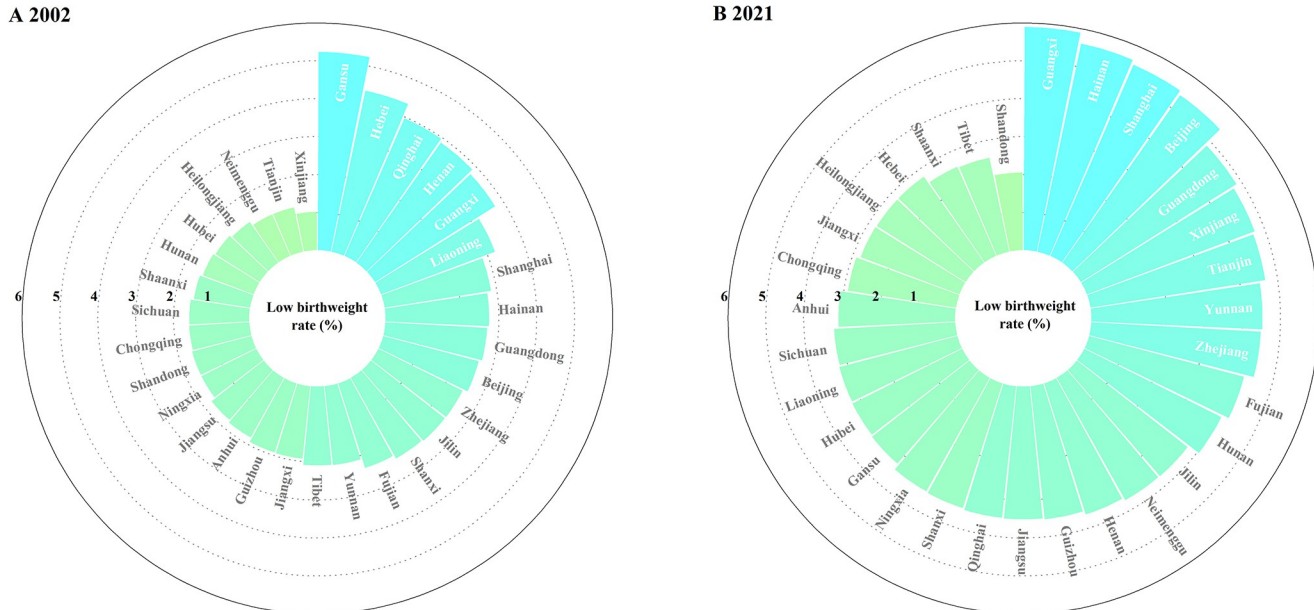

**Fig 2. Low birth weight rates and ranks in 30 provincial-level administrative districts in China, 2002 and 2021.**

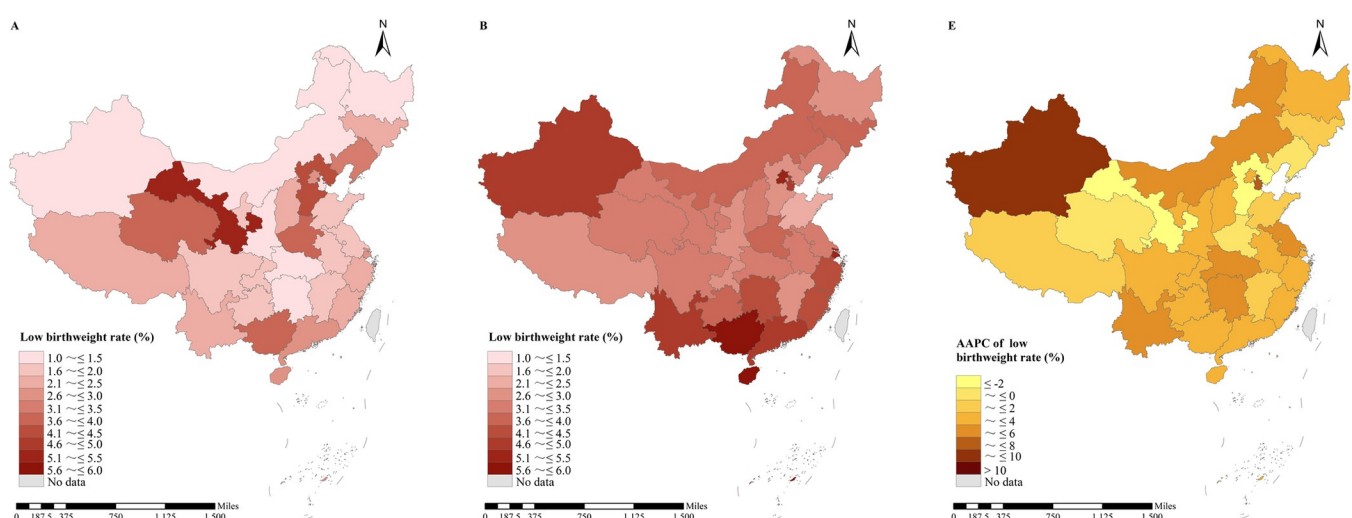

**Fig 3. Geographic distribution of the low birth weight rate and its change in China from 2002 to 2021.** (A) 1992; (B) 2021; (C) changes from 1992 to 2021. The vector maps was derived from publicly available data from the National Catalogue Service of Geographic Information in China (https://www.webmap.cn/main.do?method=index).

indicating that the higher SES group had a high LBWR. The SII and RII increased linearly over the past 20 years (SII: $B = 0.035\%$, $p < 0.001$; RII: $B = 0.011$, $p < 0.01$) (Fig 4).

## Prediction of LBWR

The parameters, Ljung-BoxQ test results and BIC values of the optimal ARIMA model are shown in S2 Table. All residual sequences were white noise sequences, and the p values of the Ljung-BoxQ test were greater than 0.05. The predicted results of the LBWR from 2022 to 2030 are shown in Fig 5. The national LBWR increased from 3.70% in 2021 to 5.28% in 2030. The LBWRs in the eastern, central, and western regions increased by 1.13%, 2.62%, and 2.02%, respectively, from 2021 to 2030. The central region (6.02%) has the highest LBWR in 2030, followed by the western (5.82%) and eastern regions (4.93%).

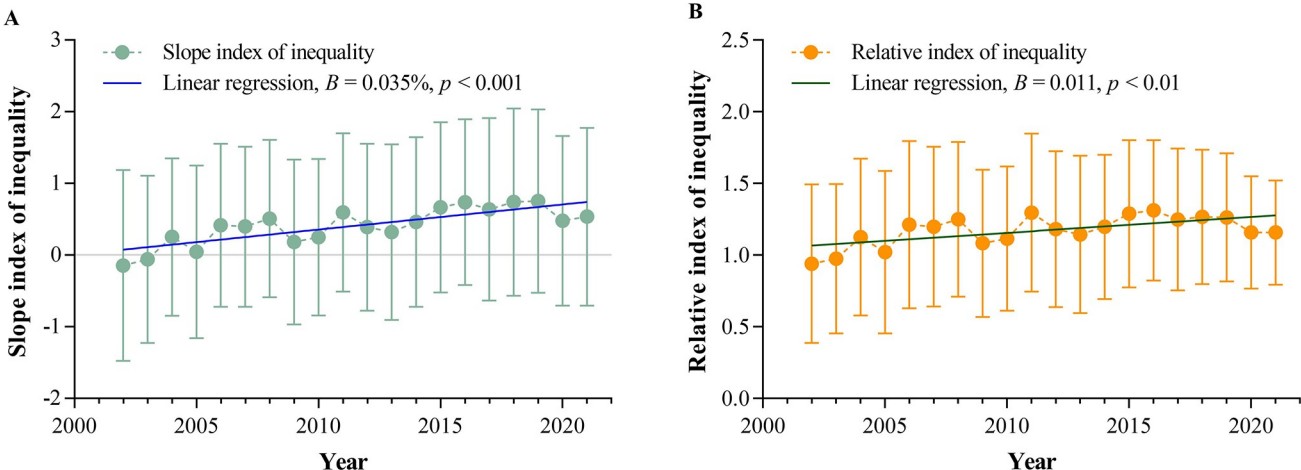

**Fig 4. Slope index of inequality and relative index of inequality of low birth weight rates in China from 2002 to 2021.** (A): slope index of inequality; (B): relative index of inequality.

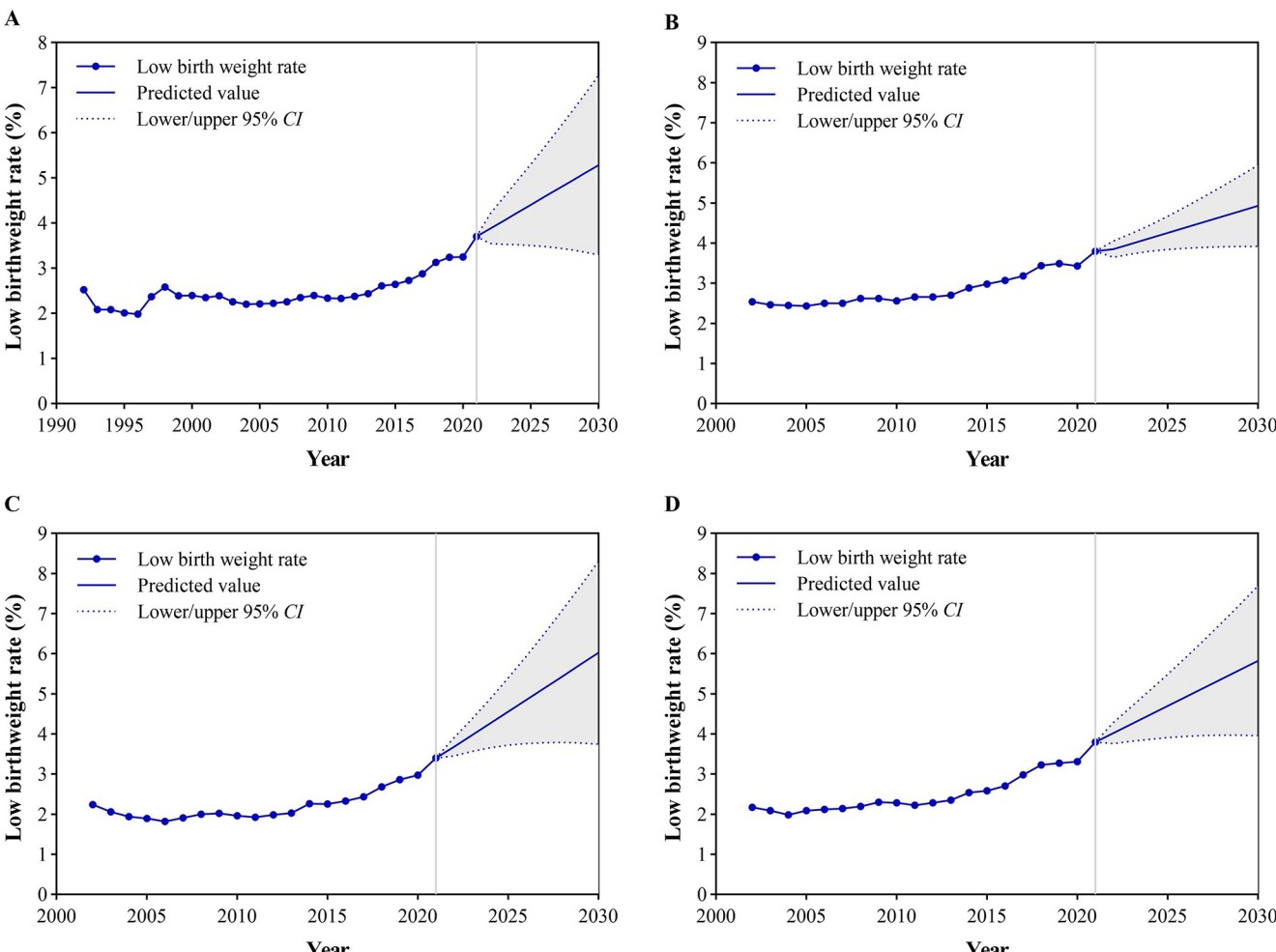

**Fig 5. The prediction of low birthweight rates from 2022 to 2030.** (A): Nation; (B): Eastern regions; (C): Central regions; (D): Western regions. The solid line represents the predictions and the gray area represents the 95% confidence intervals (CIs).

## Discussion

This paper showed that the national LBWR has continued to increase over the past 30 years. LBWR was greater in the eastern regions over the past 20 years, but in the western regions, it had the fastest increase, with provincial differences. There was socioeconomic inequality between provincial LBWRs, with low-SES groups likely to have greater LBWRs at the beginning of the 21st century. However, this relationship recently reversed, and inequality increased over the past 20 years. The national LBWR will continue to increase, exceeding 5% by 2030, and China will face greater public health problems in the future. In addition, our results reveal the provinces where maternal and child health will be vulnerable, namely, the northwest regions, areas near the capital, and some southern provinces. National interventions should prioritize these regions and allocate health, education, and medical resources appropriately.

The national LBWR has continued to increase over the past 30 years despite being at a low level in China, and this trend will hinder the achievement of the 2030 global nutrition goals. In Japan, which is also an Asian country, statistical data from the government show that LBWR was 5.1% in 1975, then gradually increased to 9.65% in 2007, and thereafter decreased to 9.41% in 2019 [15,16]. LBWR in South Korea increased from 2.6% in 1993 to 5.9% in 2016

[30]. LBWR decreased from 8.7% in 2000 to 8.2% in 2012 in eastern and southeastern Asia, but then increased to 8.8% in 2020; India has the greatest LBWR among 158 countries worldwide with 32.5% in 2000 and 32.4% in 2020 [31]. This study revealed that the APC of LBWR was 1.01% from 2004 to 2013 and 4.92% from 2013 to 2021, and the trend is similar to that of East and Southeast Asia, i.e., LBWR has risen recently. The global LBWR in 2020 was 14.7%, and Chinese LBWR was significantly lower than it [31]. Data from Peru's National Healthcare System showed that the national LBWR decreased from 6.9% in 2012 to 6.2% in 2019 [17]. Dwomoh *et al.* pooled Demographic and Health Surveys (DHS) and World Bank data from 14 West African countries from 1985 to 2019 and reported that the LBWR in West Africa was stable from 2000 to 2010 and subsequently decreased [14]. Data from the United States Centers for Disease Control and Prevention (CDC) showed a nonsignificant trend between 2003 and 2018, although the LBWR in the United States of America (USA) increased from 6.2% to 6.6% [19]. Surveys from Australia's National Perinatal Data Collection showed an increase in LBWR from 6.18% to 6.64% between 2009 and 2019 [22]. The national LBWR in Mexico increased from 6.2% in 2008 to 7.1% in 2017 [23]. In summary, there are still many countries where the LBWR has increased in the last 10 years [19,22,23,30,31]. Moreover, at the global level, the average annual rate of decline in the LBWR between 2012 and 2020 was only 0.30% per year. According to the current trend, more than 90% of the countries in the world will not be able to achieve the goal of the WHA in 2030, and according to the results of this paper, China is not an exception.

The increasing trend of LBWR was opposite to the increasing trend in maternal health care levels in China during the same period. For example, the management rate of maternity health care system, medical examination rate before birth and hospital childbirth rate increased from 65.5%, 83.7% and 60.7% in 1996 to 92.9%, 97.6% and 96.0% in 2021 in China, respectively [25]. Government investment in maternal and child healthcare (MCH) facilities has increased significantly over the past few decades, from less than 100 million yuan in 1998 to a 30-fold increase in 2017, when government investment in MCH facilities accounted for 26% of total revenue [32]. We hypothesize that although China's rapid economic development and improvements in MCH levels have reduced the risk of LBW, the accompanying changes in air pollution and the prevalence of obesity and unhealthy lifestyles could have greater negative impacts. A survey converging 54 low- and middle-income countries showed that a 10-part per billion increase in ozone concentration was associated with a 19.9-gram reduction in birth weight [33]. Another global survey revealed that the risk of LBW increased by 11% for every 10 μg/m3 increase in ambient $PM_{2.5}$, with an estimated 15.6% of LBW infants attributable to PM2.5 [34]. Although the recent trend of $PM_{2.5}$ concentrations in China was relatively stable or even slightly decreasing, ozone concentrations increased dramatically [35], and given the greater burden of disease caused by ozone compared to that caused by $PM_{2.5}$ [33,35], a part of the increase in the prevalence of LBWR may be explained by the increase in ozone concentrations. Both underweight and overweight obesity in pre-pregnant women are risk factors for LBW infants [36]. From 2004 to 2015, the prevalence of overweight and obesity in Chinese reproductive-aged women aged 15–49 years increased from 37.4% to 53.1%, with the prevalence of obesity almost doubling; sedentary time increased from 17.0 hours per week to 20.9 hours per week [37].

It is well known that older pregnant women may have more prenatal complications, which in turn are a high-risk factor for LBW [38]. In 1990, the average childbearing age of reproductive-aged women was 26.12 years, which was delayed by 2.86 years in 2020 [39]. The proportion of Chinese women giving birth over the age of 30 increased from 14.3% in 2010 to 21.1% in 2018 [40]. Deng *et al.* used data from more than 9 million Chinese maternities monitored by China's National Maternal Near Miss Surveillance System (NMNMSS) and reported that the proportion of women with prenatal complications increased by 66%, and the proportion

of women with internal diseases increased from 3.5% to 11.2% from 2012 to 2018 [41]. Previous studies have shown that alcohol consumption during pregnancy increases the risk of very low birthweight (VLBW) infants [42]. The rate of alcohol consumption among pregnant women in China increased from 2.9% in 1998 to 9.8% in 2009 [43]. The prevalence of assisted reproductive technology (ART) may also be associated with an increased risk of LBW [44]. In addition, the overall preterm birth (PTB) rate in China increased from 5.9% in 2012 to 6.4% in 2018 [41], and PTB is one of the main causes of LBWR [13].

This study also investigated the spatio-temporal characteristics of LBWR. We observed that the LBWR was greatest in the eastern regions and lowest in the central regions over the last 20 years. However, the top provinces with greater LBWRs were distributed in every area, with large differences across provinces, even within the same area. Previous studies revealed that the LBWR was greatest in the southwest regions and lowest in the central regions [24]. This may be related to the survey time, the number of provinces included, and data sources. We did find that the geographic characteristics of the LBWR changed over time. Recently, provinces with greater LBWRs initially were concentrated in the northern and southern coastal provinces. LBWR is associated with economic development, and healthcare levels vary widely across provinces in China, especially between southeastern coastal regions and remote northwestern regions [13,24]. We found that the AAPC was greatest in the western regions, and the province with the largest AAPC was Xinjiang, which was a rural western region. The group of mothers with the most disadvantaged SES was found to have the fastest increase in LBWR in Australia [22]. Interestingly, the provinces where the LBWR decreased in the last 20 years were those where the LBWR was greater in 1992, and these provinces were clustered. One possible explanation is that these provinces are adjacent to each other, have frequent interactions, and have developed interventions because of the greater LBWR. Spatio-temporal differences may also be influenced by nationality [19]. China is a developing country with a large landmass and a variety of economic and cultural combinations. A residential pattern of "mixed communities with small hives in it" was formed across nationalities. In addition, the population included most Chinese nationalities a few decades ago, while today it includes 56 nationalities (there are 56 nationalities in China) in Xinjiang. There may be complex mixed effects of these factors, and exploring the specific causes is beyond the scope of this study. However, based on the results of this study, the government can still issue a framework document of relevant policies and actions, and then each province can formulate or change relevant actions to reduce LBWR according to the local situation, such as cultural, economic, and geographic characteristics.

This study revealed greater LBWR in lower socioeconomic provinces from 2002 to 2003, followed by greater LBWR in higher socioeconomic provinces. Despite the use of different measurements, many studies have shown greater LBWRs in disadvantaged socioeconomic groups than those in advantaged socioeconomic groups [16,17,19,21–23,45,46], which is consistent with the earlier association observed in this study. Lower levels of health care, lack of knowledge about maternal health, and higher prevalence of malnutrition among pregnant women may be factors that contribute to more adverse birth outcomes in previously underdeveloped regions. However, data from Brazilian national surveys show that LBWRs were greater in developed regions (southeast and southern regions) than in underdeveloped regions (north and northeast regions) [47], and the same characteristics were found in Mexico and Pennsylvania in the USA [23,48]. The findings in these national or subnational regions are similar to our results in recent years but seem to contradict the usual results. We suggest several possible explanations for this surprising finding. First, this may be due to the poor quality of information systems in underdeveloped regions, which may lead to the underreporting of cases. Second, PTB and caesarean section (CS) rates were greater among women in developed regions [23,49]. As China's economy develops and concern for these disadvantaged economic groups, the

Chinese government has formulated and implemented many interventions, such as the "West China Development", the "Rural Revitalization Strategy", the "Targeted Poverty Alleviation", and the "Healthcare Reform". These actions promote economic development and the prioritization of labor and material resources for medical care, education and hygiene in underdeveloped regions. The living and medical standards of the population improved and nutritional needs were satisfied [32]. However, rapid economic development nationwide in China in recent years has brought new challenges such as the prevalence of overweight and obesity, severe air pollution, and unhealthy lifestyles, especially in developed provinces and urban areas.

Projections showed that LBWR would continue to increase from 2022 to 2030, especially in the central and western regions, approaching the levels of developed countries. To achieve the 2030 global nutrition targets, the WHO suggested efforts to improve maternal nutritional status and access to healthcare and hygiene at the national, community and individual levels. Our results provide baseline information for future monitoring and intervention frameworks for birth weight in China. By analyzing subnational trends over the last 20 years, we provide evidence to set realistic and tailored targets to improve recent and future birth weight indicators. Chinese national and regional governments and related departments, other developing countries, and international organizations can refer to these results to prioritize future efforts. Interestingly, the results of the provincial trends revealed that the LBWR was greater in the eastern regions than in the southern regions; however, recently, the provinces with the greatest LBWR were concentrated in the southern coastal regions near the capitals (Beijing and Tianjin) and in Xinjiang, where the fastest increase was observed. Historically, Xinjiang has been one of the regions with the lowest socioeconomic indicators and has often received interventions. Our results suggest that political and economic centers in China, southern coastal provinces with well-developed trade, and Xinjiang may need to be prioritized in the future.

As for monitoring and evaluating the problem of LBW, China has established several systems and mechanisms to ensure that the situation can be identified in time and that appropriate interventions can be taken; in the 1980s, China began to implement a rural cooperative medical insurance, which provided a health insurance for the rural population, including maternal healthcare services; in the 1990s, with the economic and social development of the country, the government paid more attention to maternal and child health and strengthened the establishment of an information system for maternal and child health; poverty alleviation is another key factor in promoting maternal and child health in China. Since 1994, China has launched a three-phase (1994–2000, 2001–2010 and 2011–2020) national poverty alleviation program that focuses on women, children and other vulnerable groups, especially those in rural and remote areas. Since 2013, the government has launched targeted poverty alleviation programs to eliminate absolute poverty by 2020, with poverty alleviation efforts targeting women and children being some of the most important priorities; in 2009, China started a new health system reform to achieve universal coverage of basic health services by expanding social and medical insurance, reforming public hospitals, and strengthening the primary health care system and public health services [32]. In 2011, China issued the "Outline of Child Development, 2011–2020", which called for controlling the LBWR to less than 4% [50]. In 2016, China issued the "National Health and Wellness Plan (2016–2020)", which called for controlling the LBWR to less than 5% [51]. In 2019, with the issuance of the "Healthy China Initiative (2019–2030)", China further emphasized improving maternal and child health [52]. However, more efforts are needed in the future to address the huge public health challenges caused by the increasing LBWR, and prioritizing interventions for poor populations or at the subnational level should be considered.

We provided the first estimates of trends in LBWR in China over the past 30 years using nationally representative authoritative data, providing evidence for global and national policy

development for the WHA's 2030 global nutrition target. We also investigated the spatio-temporal distribution and socioeconomic inequalities of LBWR over the past 20 years. This study not only provided projections of the national LBWR in China but also, for the first time, projections of the LBWR in different regions of China. Many policies and measures for LBWR interventions in China are largely based on national-level (macro) research findings and experiences and lack meso and micro evidence to support policy formulation and implementation. The information on provincial differences and spatio-temporal changes provided in this study promotes research on LBWR trends in China. It guides resource allocation and precise implementation of prevention and control measures. For other countries with rapid economic development and urbanization similar to China, the spatio-temporal characteristics identified in this study can be used as a reference for assessing local child health status. This work provides direction for the future efforts of local governments, related departments, and researchers to help improve maternal and child health as well as health equity in China. This study has several limitations. First, we obtained only provincial data for the past 20 years for analysis; earlier data were not available, and data over a longer period would help people understand national population health and regional inequalities in maternal and child health. Second, although the data are derived from national registries and are measured by professional staff in the health sector, there is no guarantee that the same measurement procedures and equipment are used at each site, which may produce some errors. For example, as the number of health facilities in each province varies, staff may have different recruitment requirements, and there may be differences in the quality of information, which particularly affects cross-province comparisons. Third, given that we used publicly available data for secondary analysis, other demographic factors that may be associated with differences in LBWR were not available. For example, the LBWR is not stratified by sex or maternal age. Concurrent trends such as overweight and obesity, family SES, and lifestyle behaviors were also not available from individuals. Nonetheless, we used the results of other studies to provide some insights into differences in the LBWR.

## Conclusion

In conclusion, the national LBWR has shown an upward trend over the past 30 years. There were large spatio-temporal differences in the LBWR in China. Over the past 20 years, the LBWR increased in most provinces but decreased in a few provinces. Although the LBWR was currently higher in the eastern regions, it increased rapidly in the western regions, and socioeconomic inequality across provinces widened. LBWR will continue to increase in the future, and based on this trend, China will not be able to achieve the 2030 global nutrition target. To address the potential increase in public health challenges in the future, it is recommended that the Chinese government increase investment in maternal and child health services, especially in the resource-poor western and northwestern regions; strengthen air quality monitoring and improvement to minimize the impact of environmental pollution on maternal and child health; promote healthy lifestyles, including sensible diets and moderate physical activity, to reduce the prevalence of overweight and obesity during pre-pregnancy. In addition, it is recommended that governments develop subnational policy and action frameworks to reduce LBWR and improve child health to ensure that global nutrition targets are met by 2030.

## Supporting information

**S1 Table. Joinpoint regression of low birthweight rate in 30 provinces of China from 2002 to 2021.** APC, annual percentage change; AAPC, average annual percentage change; CI,

confidence intervals; $^*p < 0.05$, $^{**}p < 0.01$.
(DOC)

**S2 Table. The results of the ARIMA models.** ARIMA, autoregressive integrated moving average; BIC, bayesian information criterion.
(DOC)

## Author Contributions

**Conceptualization:** Chengyue Li, Yingying Li.

**Data curation:** Chengyue Li, Lixia Lei.

**Formal analysis:** Chengyue Li, Lixia Lei.

**Funding acquisition:** Yingying Li.

**Methodology:** Chengyue Li.

**Project administration:** Yingying Li.

**Resources:** Yingying Li.

**Software:** Chengyue Li, Lixia Lei.

**Supervision:** Yingying Li.

**Validation:** Yingying Li.

**Visualization:** Lixia Lei.

**Writing – original draft:** Chengyue Li.

**Writing – review & editing:** Chengyue Li, Lixia Lei, Yingying Li.

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
