## [Decision Letter · Decision Letter 0]

27 Aug 2024

PONE-D-24-28886Spatial-temporal distribution and socioeconomic inequality of low birthweight rate in China from 1992 to 2021 and its predictions to 2030PLOS ONE

Dear Dr. LI,

Thank you for submitting your manuscript to PLOS ONE. After careful consideration, we feel that it has merit but does not fully meet PLOS ONE’s publication criteria as it currently stands. Therefore, we invite you to submit a revised version of the manuscript that addresses the points raised during the review process.

We look forward to receiving your revised manuscript.

Kind regards,

Pengpeng Ye

Academic Editor

PLOS ONE

Journal Requirements:

When submitting your revision, we need you to address these additional requirements. 1. Please ensure that your manuscript meets PLOS ONE's style requirements, including those for file naming. The PLOS ONE style templates can be found at https://journals.plos.org/plosone/s/file?id=wjVg/PLOSOne_formatting_sample_main_body.pdf and https://journals.plos.org/plosone/s/file?id=ba62/PLOSOne_formatting_sample_title_authors_affiliations.pdf 2. Please include captions for your Supporting Information files at the end of your manuscript, and update any in-text citations to match accordingly. Please see our Supporting Information guidelines for more information: http://journals.plos.org/plosone/s/supporting-information.

Reviewers' comments:

Reviewer's Responses to Questions

**Comments to the Author**

1. Is the manuscript technically sound, and do the data support the conclusions?

Reviewer #1: Yes

Reviewer #2: Yes

2. Has the statistical analysis been performed appropriately and rigorously? 

Reviewer #1: Yes

Reviewer #2: Yes

3. Have the authors made all data underlying the findings in their manuscript fully available?

Reviewer #1: Yes

Reviewer #2: Yes

4. Is the manuscript presented in an intelligible fashion and written in standard English?

Reviewer #1: Yes

Reviewer #2: Yes

5. Review Comments to the Author

Reviewer #1: 1.The manuscript is technically sound, and the data do support the conclusions.

2.The statistical analysis has been performed appropriately and rigorously.

3.The authors have made all data underlying the findings in their manuscript fully available.

4. The manuscript is presented in an intelligible fashion and written in standard English.

Reviewer #2: Reviewer’s report

Thank you for inviting me to review this important manuscript. Accept this paper with minor adjustment

Specific comments

Kindly include the keywords of this study

Discussion

Kindly explain the reason for the increment in LBWR in China.

Elaborate more and compare the findings with other Asian countries

Line 239: “…..LBWR has increased in the last 10 years”. Kindly include the reference.

Don’t just discuss the limitations of the study, also add few points regarding the strength

Highlight and discuss what the government has done so far regarding the steady rising of LBWR

Finally, add some recommendations to your conclusion

6. PLOS authors have the option to publish the peer review history of their article (what does this mean?). If published, this will include your full peer review and any attached files.

Reviewer #1: **Yes: **Kassa Demissie Abdi (PhD)

Reviewer #2: **Yes: **Segun Asaolu

---

## [Author Response · Author response to Decision Letter 0]

4 Sep 2024

Response to Reviewers

PONE-D-24-28886

Spatial-temporal distribution and socioeconomic inequality of low birthweight rate in China from 1992 to 2021 and its predictions to 2030

Reviewer 1

1.The manuscript is technically sound, and the data do support the conclusions.

2.The statistical analysis has been performed appropriately and rigorously.

3.The authors have made all data underlying the findings in their manuscript fully available.

4. The manuscript is presented in an intelligible fashion and written in standard English.

Re: Thank you for the corrections to the English phrases in the article. Keywords have been added and references have been checked. See Tracking Changes manuscript for other revisions.

Reviewer 2

Kindly include the keywords of this study

Re: Add keywords.

Discussion

Kindly explain the reason for the increment in LBWR in China.

Re: The reasons for the increase in LBWR are discussed considerably in paragraphs 3-4 of the Discussion section, with possible factors being the increase in ozone concentration, the increase in the prevalence of overweight and obesity among pregnancy-age women, the delay in the reproductive age of females, the increase in prenatal complications, the increase in the proportion of alcohol consumption among pregnant women, etc. Pages 7-8, lines 267-304.

Elaborate more and compare the findings with other Asian countries 

Re: Added reference 31, which is the most recently published evidence of global trends in LBWR, including most countries worldwide (also including Asia). Unfortunately, except for the data from reference 31 and Korea and Japan provided in the original article, there is little evidence of recent trends from Asia. Page 7, lines 247-251.

Line 239: “…..LBWR has increased in the last 10 years”. Kindly include the reference. 

Re: Added references “[19,22-23,30-31]”. Page 7, line 263.

Don’t just discuss the limitations of the study, also add few points regarding the strength

Re: The strengths of this study were added at the end of the discussion. Pages 10-11, lines 397-410.

Highlight and discuss what the government has done so far regarding the steady rising of LBWR

Re: A related discussion has been added to the penultimate paragraph of the Discussion section. Page 10, lines 374-396.

Finally, add some recommendations to your conclusion

Re: Some suggestions have been added to the conclusion section. Page 11, lines 433-441.

---

## [Decision Letter · Decision Letter 1]

4 Oct 2024

PONE-D-24-28886R1

Spatio-temporal distribution and socioeconomic inequality of low birthweight rate in China from 1992 to 2021 and its predictions to 2030

PLOS ONE

Dear Dr. Li, I am writing with an update about your submission, "Spatio-temporal distribution and socioeconomic inequality of low birthweight rate in China from 1992 to 2021 and its predictions to 2030" (PONE-D-24-28886).  After this article received an Accept decision you requested several  authorship changes.  The nature, extent and timing of the requests to change the author list call into question whether your manuscript complies with the PLOS Authorship policy (https://journals.plos.org/plosone/s/authorship). As such, we are rescinding the Accept decision and rejecting the manuscript.  We may reconsider this submission in the future if a research integrity official at the corresponding author’s institution reviews and provides verification of the article's authors and their contributions. The following documents would need to be provided when resubmitting:a.     
Written, signed statements from all contributors, including added/removed authors, confirming that all agree with the article’s author list and contributionsb.     
Cover letter that describes the contribution of each author and provides a specific reason why each author was added or removed after initial submissionc.     
Formal letter from a research integrity official or equivalent at the corresponding author’s institution, or the institution where the majority of the research was conducted, confirming the author list and stated contributions.d.     
Institutional email address for the official responsible for oversight of research and/or research integrity at the corresponding author’s institution.  I am sorry we do not have more positive news, but hope that you understand the reasons why we rejected this submission. Kind regards,Jennifer Tucker, PhD

Staff Editor

PLOS ONE

Additional Editor Comments (if provided):

Reviewers' comments:

Reviewer's Responses to Questions**Comments to the Author**

1. If the authors have adequately addressed your comments raised in a previous round of review and you feel that this manuscript is now acceptable for publication, you may indicate that here to bypass the “Comments to the Author” section, enter your conflict of interest statement in the “Confidential to Editor” section, and submit your "Accept" recommendation.Reviewer #1: All comments have been addressedReviewer #2: All comments have been addressed**********2. Is the manuscript technically sound, and do the data support the conclusions?

The manuscript must describe a technically sound piece of scientific research with data that supports the conclusions. Experiments must have been conducted rigorously, with appropriate controls, replication, and sample sizes. The conclusions must be drawn appropriately based on the data presented. Reviewer #1: YesReviewer #2: Yes**********3. Has the statistical analysis been performed appropriately and rigorously? Reviewer #1: YesReviewer #2: Yes**********4. Have the authors made all data underlying the findings in their manuscript fully available?

The PLOS Data policy requires authors to make all data underlying the findings described in their manuscript fully available without restriction, with rare exception (please refer to the Data Availability Statement in the manuscript PDF file). The data should be provided as part of the manuscript or its supporting information, or deposited to a public repository. For example, in addition to summary statistics, the data points behind means, medians and variance measures should be available. If there are restrictions on publicly sharing data—e.g. participant privacy or use of data from a third party—those must be specified.Reviewer #1: YesReviewer #2: Yes**********5. Is the manuscript presented in an intelligible fashion and written in standard English?

PLOS ONE does not copyedit accepted manuscripts, so the language in submitted articles must be clear, correct, and unambiguous. Any typographical or grammatical errors should be corrected at revision, so please note any specific errors here.Reviewer #1: YesReviewer #2: Yes**********6. Review Comments to the Author

Please use the space provided to explain your answers to the questions above. You may also include additional comments for the author, including concerns about dual publication, research ethics, or publication ethics. (Please upload your review as an attachment if it exceeds 20,000 characters)Reviewer #1: 1. All comments have been addressed.

2. The manuscript is technically sound, and the data do support the conclusions.

3. The statistical analysis has been performed appropriately and rigorously.

4. The authors have made all data underlying the findings in their manuscript fully available.

5. The manuscript is presented in an intelligible fashion and written in standard English.Reviewer #2: The author has answered all the questions and comments accordingly, and the manuscript is now ready for publication. Please Kindly Accept this paper**********7. PLOS authors have the option to publish the peer review history of their article (what does this mean?). If published, this will include your full peer review and any attached files.

Reviewer #1: **Yes: **Kassa Demissie Abdi (PhD)Reviewer #2: **Yes: **Segun Asaolu********** 

- - - - -

---

## [Author Response · Author response to Decision Letter 1]

23 Oct 2024

PONE-D-24-28886

Spatial-temporal distribution and socioeconomic inequality of low birthweight rate in China from 1992 to 2021 and its predictions to 2030

Reviewer 1

1.The manuscript is technically sound, and the data do support the conclusions.

2.The statistical analysis has been performed appropriately and rigorously.

3.The authors have made all data underlying the findings in their manuscript fully available.

4. The manuscript is presented in an intelligible fashion and written in standard English.

Re: Thank you for the corrections to the English phrases in the article. Keywords have been added and references have been checked. See Tracking Changes manuscript for other revisions.

Reviewer 2

Kindly include the keywords of this study

Re: Add keywords.

Discussion

Kindly explain the reason for the increment in LBWR in China.

Re: The reasons for the increase in LBWR are discussed considerably in paragraphs 3-4 of the Discussion section, with possible factors being the increase in ozone concentration, the increase in the prevalence of overweight and obesity among pregnancy-age women, the delay in the reproductive age of females, the increase in prenatal complications, the increase in the proportion of alcohol consumption among pregnant women, etc. Pages 7-8, lines 267-304.

Elaborate more and compare the findings with other Asian countries 

Re: Added reference 31, which is the most recently published evidence of global trends in LBWR, including most countries worldwide (also including Asia). Unfortunately, except for the data from reference 31 and Korea and Japan provided in the original article, there is little evidence of recent trends from Asia. Page 7, lines 247-251.

Line 239: “…..LBWR has increased in the last 10 years”. Kindly include the reference. 

Re: Added references “[19,22-23,30-31]”. Page 7, line 263.

Don’t just discuss the limitations of the study, also add few points regarding the strength

Re: The strengths of this study were added at the end of the discussion. Pages 10-11, lines 397-410.

Highlight and discuss what the government has done so far regarding the steady rising of LBWR

Re: A related discussion has been added to the penultimate paragraph of the Discussion section. Page 10, lines 374-396.

Finally, add some recommendations to your conclusion

Re: Some suggestions have been added to the conclusion section. Page 11, lines 433-441.

JOURNAL REQUIREMENTS:

1. Thank you for sharing that the vector maps were derived from publicly available data from the National Catalogue Service of Geographic Information in China (https://www.webmap.cn/main.do?method=index). We request the authors to attribute the source of the basemap in the corresponding figure legend. As with all content, we ask that authors respect map providers’ requirements for attribution.

Re: modified.

The nature, extent and timing of the requests to change the author list call into question whether your manuscript complies with the PLOS Authorship policy (https://journals.plos.org/plosone/s/authorship). As such, we are rescinding the Accept decision and rejecting the manuscript. 

We may reconsider this submission in the future if a research integrity official at the corresponding author’s institution reviews and provides verification of the article's authors and their contributions. The following documents would need to be provided when resubmitting:

a. Written, signed statements from all contributors, including added/removed authors, confirming that all agree with the article’s author list and contributions

b. Cover letter that describes the contribution of each author and provides a specific reason why each author was added or removed after initial submission

c. Formal letter from a research integrity official or equivalent at the corresponding author’s institution, or the institution where the majority of the research was conducted, confirming the author list and stated contributions.

d. Institutional email address for the official responsible for oversight of research and/or research integrity at the corresponding author’s institution.

Re: We have submitted these documents.

---

## [Editor Report · Decision Letter 2]

1 Dec 2024

Spatio-temporal distribution and socioeconomic inequality of low birthweight rate in China from 1992 to 2021 and its predictions to 2030

PONE-D-24-28886R2

Dear Dr. Li,

We’re pleased to inform you that your manuscript has been judged scientifically suitable for publication and will be formally accepted for publication once it meets all outstanding technical requirements.

Kind regards,

Jennifer Tucker, PhD

Staff Editor

PLOS ONE
---

## [Editor Report · Acceptance letter]

27 Dec 2024

PONE-D-24-28886R2 

PLOS ONE

Dear Dr. Li, 

I'm pleased to inform you that your manuscript has been deemed suitable for publication in PLOS ONE. Congratulations! Your manuscript is now being handed over to our production team.

Kind regards, 

on behalf of

Dr Jennifer Tucker 

Staff Editor

PLOS ONE